# The Riddle of the Double Vision—A Rare Case of Intracranial Tumor: When Imaging Resolves the Mystery

**DOI:** 10.3390/diagnostics14090932

**Published:** 2024-04-29

**Authors:** Radina Kirkova, Svetla Dineva, Elisa Stradiotto, Ivan Tanev, Alessandra Di Maria

**Affiliations:** 1Department of Ophthalmology, Vita-Salute San Raffaele University, 20132 Milan, Italy; 2Eye Clinic Zrenieto, 1000 Sofia, Bulgaria; itanev@skycode.com; 3Diagnostic Imaging Department, National Cardiology Hospital, 1000 Sofia, Bulgaria; svetladineva7@gmail.com; 4Imaging Department, Medical University of Sofia, 1000 Sofia, Bulgaria; 5Department of Biomedical Sciences, Humanitas University, 20090 Milan, Italy; elisa.stradiotto94@gmail.com; 6Department of Ophthalmology, IRCCS Humanitas Research Hospital, 20089 Milan, Italy; alessandra.di_maria@humanitas.it

**Keywords:** sinonasal, squamocellular, carcinoma, sinusitis, double vision, ophthalmic, oral

## Abstract

A 77-year-old-man with arterial hypertension, diabetes mellitus type II presented at our clinic for a routine ophthalmological exam. He complained of intermittent double vision. The ophthalmic examination revealed paralysis of III (n. oculomotorius) and VI (n. abducens) cranial nerves with ptosis, deficit in elevation and abduction of the left eye. The patient underwent urgent MRI imaging of the brain/orbits and paranasal sinuses, and urgent neurological assessment. MRI revealed a volume-occupying process, starting from the posterior wall of the left maxillary sinus with perineural diffusion and involvement of the homolateral trigeminal nerve, intracranial spread in the medial cranial fossa and involvement of the cavernous, sphenoidal sinuses and the orbital apex on the left side. Biopsy was performed, and the histology resulted in sinonasal squamous cell carcinoma with intracranial spread.

A 77-year-old-man with arterial hypertension, diabetes mellitus type II presented at our clinic for a routine ophthalmological exam. He reported a history of a 5-month-old left maxillary sinusitis and complained of intermittent double vision for 2 weeks. From a general health point of view, he suffered from a significant weight loss in the last month. There was no history of trauma.

The best corrected visual acuity was 0.95 in the right eye and 0.7 in his left eye.

The ophthalmic examination revealed paralysis of III (n. oculomotorius) and VI (n. abducens) cranial nerves with ptosis, deficit in elevation and abduction of the left eye. The extrinsic ocular motility of the right eye was normal. The anterior segment of both eyes showed no significant alterations, except for early-stage cataracts without particular relevance in relationship to his age. It is of note that the pupillary reflexes were preserved. The posterior segment examination did not display any sign of diabetic retinopathy, nor were any other retinal alterations found.

In light of the neurological symptoms, we requested urgent computed axial tomography (CT), followed by magnetic resonance imaging (MRI) of the encephalon and the orbits. The systemic evaluation was completed by routine blood tests and by neurologic, ENT, and endocrinologic visits.

The CT and MRI (Figure 1) revealed a volume-occupying process, from the posterior wall of the left maxillary sinus with perineural infiltration and involvement of the homolateral trigeminal nerve, intracranial spread in medial cranial fossa and involvement of the cavernous, sphenoidal sinuses and the orbital apex on the left side. The imaging raises a serious suspicion of a more distant origin of the malignant process (Figure 2A—tongue, indicated with a blue arrow), which after a tooth extraction can easily spread to the maxillary sinus.

For precision of the diagnosis, baseline staging of the process and planning of the therapeutic approach an FDG-PET/CT scan was required (Figure 4). FDG-PET/CT has the advantage of assessing morphologic as well as molecular information on cellular disease activity. It is also useful in detecting the primary lesion in tumors of unknown origin. In our case, the FDG-PET/CT did not reveal extension of the malign process outside the borders defined with the other imaging modalities, and the results of all imaging modalities were compatible.

In order to obtain the histological diagnosis, an endonasal biopsy was performed, which resulted in sinonasal squamous cell carcinoma.

Sinonasal SCC (SNSCC) is a rare, aggressive malignancy that represents 0.2% of the malignant tumors in the human body [1]. It is believed to arise from the Schneiderian membrane, that covers the sinuses and the nasal cavity, and it mostly affects male patients in the sixth decade of life [2]. According to the American Cancer Society, the risk factors associated with the disease include human papilloma virus infection (HPV), smoking, and occupational risks such dust from textiles, leather and woods, flour, nickel and chromium exposure, glues, formaldehyde, organic solvents, mustard gas, and radiation. HPV-positive cases are associated with patients of a younger age (58 years) [3]. SCC has the tendency to mimic various banal conditions in its initial stages (such as sinusitis), and a relatively high number (up to 12%) of the patients are asymptomatic until the late stages of the disease [4]. In our case, the process started mimicking left maxillary sinusitis. External ocular motility is rarely affected in the course of maxillary sinusitis.

Since our patient has diabetes mellitus, it was crucial to make a differential diagnosis with cranial nerve palsy due to diabetic microangiopathy. Glucose blood levels were within normal limits; the endocrinologist confirmed the lack of alterations.

After the CT was performed, it revealed a volume-occupying lesion in the left maxillary sinus with perineural spread and invasion in the orbit, other sinuses and medial cranial fossa. Sinusitis does not have invasive characteristics, so it was excluded as a diagnostic option. In such cases an endonasal biopsy is the key point to the diagnosis. In our case, the histological result was classified as squamous cell carcinoma G1 (Figure 2).

SCC of the central nervous system is usually due to the invasion of a primary tumor of the head–neck area. Primary intracranial squamous cell carcinoma (PISCC) is extremely uncommon, and only nine cases have been reported in the literature up to date [5]. When primary, it usually arises from a benign epidermoid or dermoid cyst [6]. In our case, we possess CT images made six months prior to the patient’s first visit with us, in which it was described only as “left maxillary sinusitis”, without a description of a dermoid cyst. As it does not meet the criteria of Garcia and Hamlat for malignant transformation, we believe that the primary lesion was in the maxillary sinus [7].

The local invasion of the tumor is well registered chronologically on the CT scans and demonstrates the aggressivity and fast progression of the tumor.

Endonasal endoscopic biopsy was performed. The histology resulted in squamous cell carcinoma G1.

A “gold” standard for the treatment of SCC with intracranial invasion is up to be defined. It depends on the complexity of the case and includes surgery, chemotherapy, or radiotherapy.

The patient underwent an FDG-PET total body scan, MRI, echography of local lymph nodes and a multidisciplinary consultation to establish a therapeutic strategy. FDG-PET is an extremely useful and precious method in determining the systemic extension of the process and has a crucial role in the planning of the therapeutic approach if systemic metastases are detected. The method has a higher diagnostic performance compared to CT and MRI alone. Since the tumor process in our case was very invasive with distorted normal structures, the interpretation of CT and MRI scans was quite problematic. The FDG-PET/CT was extremely useful in defining the exact borders of the process and searching of distant metastases. There were no metastases and no involvement of the lymph nodes. Because of the large dimensions of the tumor process, its proximity to vital centers, and difficult surgery approach, a decision for performing radiotherapy was made. The patient received treatment doses of 12 Gy. Nagasawa et al. reported a 29.2-month average survival rate in patients with advanced intracranial SCC, who underwent radiotherapy with doses varying from 12 Gy to 15 Gy [8].

After 11 cycles of radiotherapy and a follow-up period of 11 months (MRI, PET), the tumor process is stable with no extension, no metastasis, and no progression of the functional deficits.

SCC with such an impressive volume-occupying intracranial spread is a rare clinical finding. The diagnosis is often a mystery, and the ophthalmologist must remember that every hint of variation in the neuro-ophthalmological examination leads to the necessity of CT or MRI. FDG-PET/CT has a unique role in staging, therapy planning, and following-up in neuro-oncological cases. It is a highly informative and rapidly expanding imaging modality because of the innovation in radiopharmaceuticals, which allow diagnostic precision and better tumor coverage (for example, radiotracers for tumor perfusion, angiogenesis, neuroinflammation, etc.). Managing patients like these is a challenge for clinicians and requires a multi-disciplinary approach, because of the wide extension of the process and the involvement of various structures from different competences and vital significance. Treating every single patient depends on the individual characteristics of the malignant process and it is a state of the art; very often, a combination of therapeutic strategies is needed when possible (chemotherapy, radiotherapy, surgery, and gamma-knife radiosurgery).

## Figures and Tables

**Figure 1 diagnostics-14-00932-f001:**
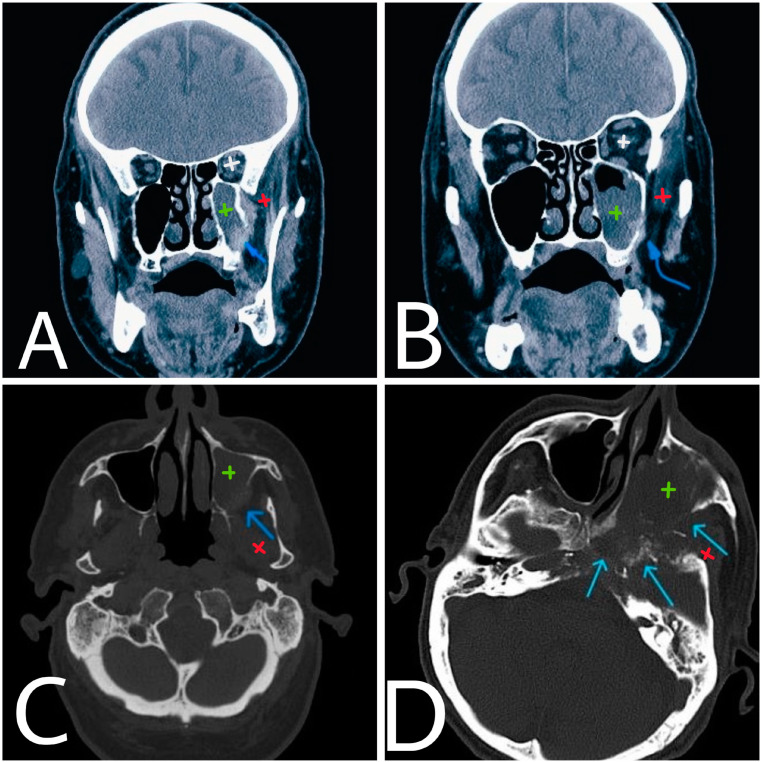
Coronal and axial CT images at the level of maxillary sinuses (**A**,**B**). The lateral wall of maxillary sinus is presented with local osteolytic defect (blue arrow) and infiltration of adjacent soft tissue structures (**C**). 12 months later (**D**), the process has advanced with extensive bone destruction of a larger area (arrows). White—left orbit. Green—left maxillary sinus. Red—left masticator space.

**Figure 2 diagnostics-14-00932-f002:**
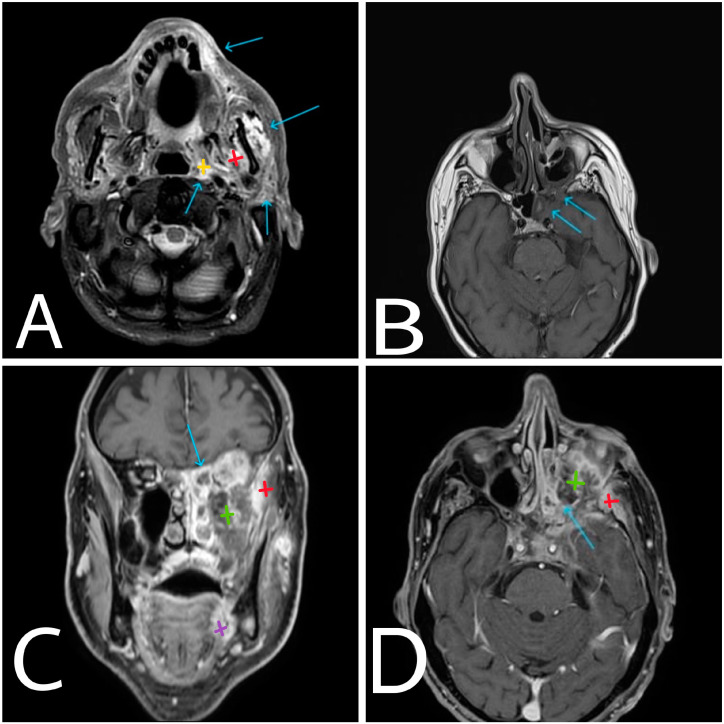
Axial post-contrast enhanced T1-weighted fat-saturated (fs) MRI (**A**) at the basis of the maxillary sinuses where extensive surrounding tissue involvement is present (arrows). Axial contrast-enhanced T1-weighted (**B**) and coronal and axial contrast-enhanced T1-weighted fat-saturated (**C**,**D**) MRI images of the affected area on the left. Arrows are pointing at the area of intracranial involvement of mid-cerebral fossa. Yellow—left parapharyngeal space. Green—left maxillary sinus. Red—left masticator space. Purple—highly suspicious area for primary neoplasm at the level of left lingual surface.

**Figure 3 diagnostics-14-00932-f003:**
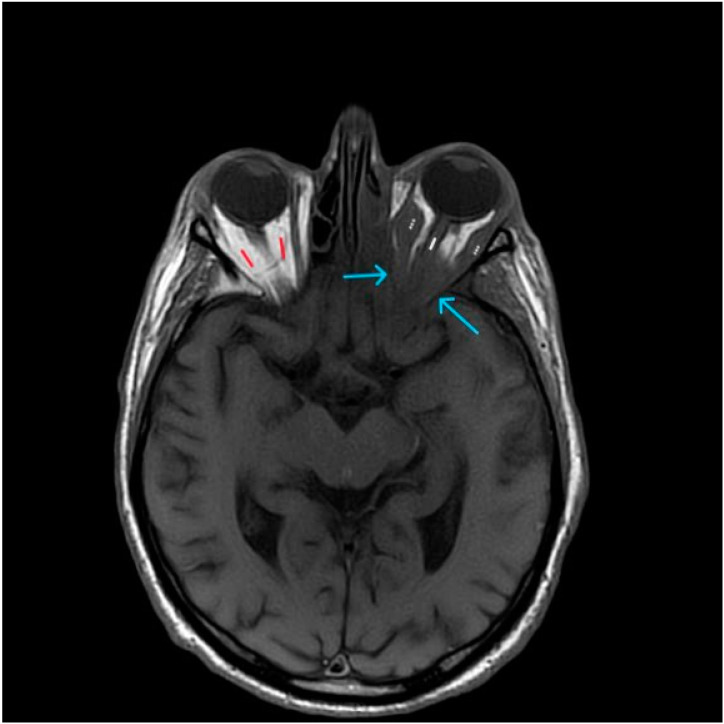
Axial T1-weighted image at the level of the orbits. Arrows are pointing at the reduced intraconal fat tissue due to intracranial extension of the pathological process in the left maxillary sinus. White line—optic nerve. Dots—medial and lateral oculomotor muscles. Red—normal hyperintense (white) fat tissue of the intact orbit.

**Figure 4 diagnostics-14-00932-f004:**
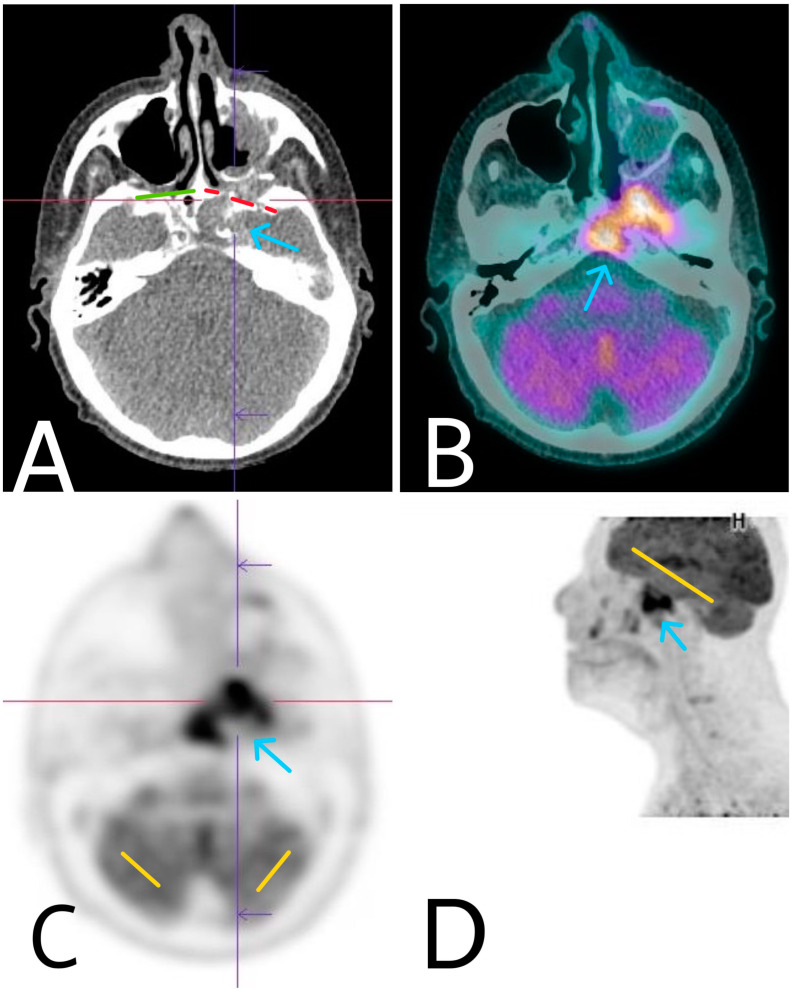
Axial CT (**A**), axial (**C**), and sagittal (**D**) FDG-PET and axial-fused PET-CT (**B**) image showing higher tumor metabolic activity, compared to the brains physiological one, extensive bone destruction, and extensive intracranial involvement by the primary pathological process. Green—intact bony borders of right orbit. Red—lytic bony borders of infiltrated left orbit. Yellow—normal brain metabolic activity.

**Figure 5 diagnostics-14-00932-f005:**
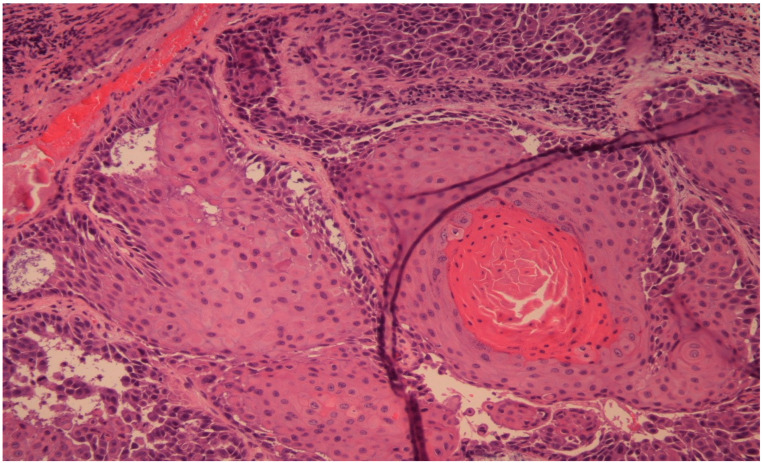
Hematoxylin and eosin staining, original magnification ×40. The histology result confirms SCC with focal acantholytic aspects and hyperkeratosis.

## Data Availability

Data are available upon reasonable request to the corresponding author.

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
