# Peer review of "The Riddle of the Double Vision—A Rare Case of Intracranial Tumor: When Imaging Resolves the Mystery"

_diagnostics, 2024, doi:10.3390/diagnostics14090932_

Round 1

Reviewer 1 Report

Comments and Suggestions for Authors

Dear authors,

This case presents a 77-year-old man with arterial hypertension and diabetes mellitus type II, who sought a routine ophthalmological exam due to intermittent double vision. The subsequent findings of cranial nerve paralysis, specifically III and VI, and the subsequent diagnosis of sinonasal squamous cell carcinoma with intracranial spread are concerning and highlight the interconnectedness of various medical disciplines.

From a critical perspective, there are various factors that deserve attention:

1.     Imaging technology and its limitations: While the MRI revealed a volume-occupying process with intracranial spread, a critical perspective could discuss the limitations of imaging technology. Are there advancements in imaging that could enhance early detection or provide more detailed information about the extent of the tumor? Examining the present condition of imaging technology and its prospective advancements could enhance the process of deeply assessing it.

2.     Kindly furnish additional MRI images in accordance with the specified procedure, including T1 weighted, T2 weighted, FLAIR, diffusion-weighted imaging (DWI), SWI, postcontrast sequences, susceptibility weighted imaging (SWI), perfusion, and spectroscopy. Please also offer a detailed description of each image.

3.     Kindly submit images depicting the MRI and PET-CT aspects during the 11-month period of radiation and follow-up.

4.     A careful examination of the content and correction of certain words and typographical errors would be required.

In conclusion, while the article provides a detailed clinical case, a critical perspective could involve examining the efficiency of the diagnostic pathway, the limitations of current medical technologies, the effectiveness of multidisciplinary collaboration, the role of patient education, and the holistic aspects of patient-centered care.

Comments on the Quality of English Language

A careful examination of the content and correction of certain words and typographical errors would be required.

Author Response

Dear Reviewer,

Thank you for dedicating your time in reviewing our manuscript and four the detailed review.

We tried to satisfy all of the four remarks. We added more images and improved the imaging description section

Reviewer 2 Report

Comments and Suggestions for Authors

The authors describe a case of sinonasal squamous cell carcinoma which presented with
ophthalmoplegia. Imaging was appropriately performed to identify the structural abnormality, which
was then appropriately biopsied. The malignancy was treated with radiation therapy.

The single FIGURE included should have labels and a detailed legend to direct the reader to the portions of the image which make it interesting. A coronal view in addition to the axial view would make the presentation for interesting, as well.
The portions of the text in the DISCUSSION which describe the case should be moved to the Case Report section. The DISCUSSION could be shortened.

Comments on the Quality of English Language

English language editing is needed for spelling and sentence structure. For example, in the TITLE, "Mystery" is misspelled.

Author Response

Dear Reviewer,

Thank you for the review.

We tried to satisfy all of your remarks - we added more images and description.

Round 2

Reviewer 1 Report

Comments and Suggestions for Authors

Dear authors,

The revised version of the manuscript "THE RIDDLE OF THE DOUBLE VISION – A RARE CASE OF INTRACRANIAL TUMOR: WHEN IMAGING RESOLVES THE MYSTERY" presents a 77-year-old man with arterial hypertension and diabetes mellitus type II, who sought a routine ophthalmological exam due to intermittent double vision. The subsequent findings of cranial nerve paralysis, specifically III and VI, and the subsequent diagnosis of sinonasal squamous cell carcinoma with intracranial spread are concerning and highlight the interconnectedness of various medical disciplines. 

1. The images provided comply with my requirements.

2. A careful examination of the content and correction of certain words and typographical errors would be required.

3. Please provide a conclusion or final statement for the manuscript.

Comments on the Quality of English Language

A careful examination of the content and correction of certain words and typographical errors would be required.

Author Response

Dear Reviewer,

The text was corrected carefully by a native English speaker.

We provided a conclusion

Hope we satisfied your remarks

Reviewer 2 Report

Comments and Suggestions for Authors

The authors have addressed some of the reviewer's comments. Labeling of key normal anatomic structures would make the report more useful and more interesting.

Comments on the Quality of English Language

English language needs moderate editing

Author Response

Dear Reviewer

We provided labeling of key normal anatomic structures.

The English editing was made by native English speaker

All of the corrections are highlighted in red